# A Bayesian method to infer copy number clones from single-cell RNA and ATAC sequencing

**Lucrezia Patruno**[1,2©¤], **Salvatore Milite**[2,3©], **Riccardo Bergamin**[2], **Nicola Calonaci**[2], **Alberto D'Onofrio**[2], **Fabio Anselmi**[2], **Marco Antoniotti**[1,4], **Alex Graudenzi**[1,4], **Giulio Caravagna**[2]*

**1** Department of Informatics, Systems and Communication, Università degli Studi di Milano-Bicocca, Milan, Italy, **2** Department of Mathematics and Geosciences, Università degli Studi di Trieste, Trieste, Italy, **3** Centre for Computational Biology, Human Technopole, Milan, Italy, **4** B4—Bicocca Bioinformatics Biostatistics and Bioimaging Centre, Università degli Studi di Milano-Bicocca, Milan, Italy

☯ These authors contributed equally to this work.
¤ Current address: University College London Cancer Institute, London, United Kingdom
* gcaravagna@units.it

**Data Availability Statement:** Following the predecessor CONGAS, CONGAS+ is developed as two open source packages (one written in Python and another in R) available on GitHub. This release

## Abstract

Single-cell RNA and ATAC sequencing technologies enable the examination of gene expression and chromatin accessibility in individual cells, providing insights into cellular phenotypes. In cancer research, it is important to consistently analyze these states within an evolutionary context on genetic clones. Here we present CONGAS+, a Bayesian model to map single-cell RNA and ATAC profiles onto the latent space of copy number clones. CONGAS+ clusters cells into tumour subclones with similar ploidy, rendering straightforward to compare their expression and chromatin profiles. The framework, implemented on GPU and tested on real and simulated data, scales to analyse seamlessly thousands of cells, demonstrating better performance than single-molecule models, and supporting new multi-omics assays. In prostate cancer, lymphoma and basal cell carcinoma, CONGAS+ successfully identifies complex subclonal architectures while providing a coherent mapping between ATAC and RNA, facilitating the study of genotype-phenotype maps and their connection to genomic instability.

## Author summary

Cancer cells, compared to normal ones, often acquire or lose chromosomes and, in the context of cancer progression and treatment response, copy number alterations can drive subclonal cancer dynamics. Here, we seek to use single-cell sequencing data from both RNA and ATAC data to determine tumour subclones associated with copy number alterations. We develop a Bayesian model that links a latent copy number state per subclone with transcript abundance (RNA) and chromatin accessibility (ATAC), and can cluster cells into subclones with similar copy number profiles. Our model works when input cells are sequenced independently, or with modern multi-omics protocols. By linking

of CONGAS+ substitutes the earlier CONGAS, and is provided as 1) a Python implementation for the inference models, available at https://github.com/caravagnalab/CONGASp 2) an R implementation for data processing and visualisation functions, available at https://github.com/caravagnalab/rcongas The tool comes with example data, vignettes for basic analyses and an online manual reachable from https://caravagnalab.github.io/rcongas/. The BCC scRNA and scATAC dataset were downloaded from Gene Expression Omnibus (GEO), under accession numbers GSE123814 and GSE129785 respectively. BCC whole exome sequencing data was downloaded from SRA under accession number PRJNA533341. scRNA data for the gastric cell line was obtained from GEO under accession GSE142750, and scATAC was downloaded from SRA under accession PRJNA674903. 10x multiome data for the B-cell lymphoma was accessed from the 10x genomics website. scRNA and scATAC data for the prostate cancer cell line was downloaded from GEO (GSE168669). The code to reproduce the analyses in the text, as well as the data, is available at the GitHub repository https://github.com/caravagnalab/CONGASp_supplementary_data.

**Funding:** This work was funded by the CRUK/AIRC Accelerator Award #22790 to (MA, AG and GC), "Single-cell Cancer Evolution in the Clinic", by AIRC under MFAG 2020 - ID. 24913 project – P.I. Caravagna Giulio to (GC), by the European Commission Program PPPA2027, PPPA-2021-AIPC #LC-01815952/101052609 to (MA) and by the 2021 FAQC program of the Universitá degli Studi di Milano-Bicocca to (MA). The funders had no role in study design, data collection and analysis, decision to publish, or preparation of the manuscript.

**Competing interests:** The authors have declared that no competing interests exist.

aneuploidy to gene expression and chromatin conformation, our approach provides a novel way to map genotypes with phenotypes to study the molecular basis of cancer heterogeneity.

## Introduction

Cancer is a disease where multiple sub-clones evolve under positive, neutral and negative selection forces, with the complex molecular configuration of each sub-clone determining selection dynamics [1–3]. We can probe a number of such configurations using modern sequencing assays—from DNA-seq we measure the genome, from RNA-seq the transcriptome and from ATAC-seq the epigenome—which, in these days, are increasingly applied at the single-cell level. Moreover, with the recent adoption of multiomics assays, we can also probe different molecules (e.g., RNA and ATAC) from the same cell [4–6]. Single-cell data, combined with the development of patient-derived model systems [7], opens the possibility of studying complex subclonal configurations across different molecules, with key translational repercussions [8].

From a computational perspective, this type of analysis has sparkled a number number of important data integration challenges (see the review in [9]). Extending earlier works of [10], in this paper we approach a problem of unsupervised diagonal integration [11], mapping single-cell RNA (scRNA-seq) and ATAC (scATAC-seq) data of independent or multiomics assays in a biologically-interpretable latent space. Contrary to biology-unaware models such as variational autoencoders or factor analysis [12–15], we link RNA and ATAC on the same DNA configuration because RNAs are produced by transcription of a DNA template, and ATAC is an assessment of chromatin conformation, a physical feature of the molecule. Reasoning that both the transcriptional activity of a gene, as well as the amount of open chromatin are influenced by the number of DNA copies of a gene, we attempt at determining the latent copy number profile of a cell as a DNA feature [10]. Therefore, we map RNA and ATAC measurements on a common latent space reflecting the number of DNA copies, a proxy for total copy numbers, while clustering cells into subclones with similar profiles.

Copy number inference algorithms for scRNA-seq or scATAC-seq have been independently investigated for both molecules [16–22]. For scRNA-seq, we recently introduced CONGAS [10], a probabilistic method to infer copy number subclones. CONGAS improved over competing methods that decoupled copy number inference from clustering and, in this work, we follow a similar rationale to develop CONGAS+, which augments scRNA-seq with scATAC-seq data in order to resist statistical confounders such as allele-specific expression and post-transcriptional regulation [23–25]. CONGAS+ can integrate independent/multiomics scRNA-seq/scATAC-seq measurements using a biology-informed latent space model. Using probabilistic programming on GPUs [26] as in CONGAS, stochastic variational inference and gradient descent allow CONGAS+ to analyse tens of thousands of cells seamlessly. In this paper, we prove this by analysing a B-cell lymphoma ($\sim 6400$), a basal cell carcinoma ($\sim 2400$ cells), a prostate cancer cell line ($\sim 16000$ cells) and a gastric cell line ($\sim 7000$ cells) in various combinations of scRNA-seq and scATAC-seq assays.

## Materials and methods

### The CONGAS+ statistical model

CONGAS+ is a Bayesian model to infer and cluster, from scRNA-seq and scATAC-seq of independents or multiomics assays, phylogenetically related clones with distinct Copy Number

Alterations (CNAs, Fig 1A). Clones distinguished by point mutations, which are often those used to determine clonal evolution patterns, cannot be identified by this approach. However, CNA-associated clones can still be linked to selection and associated to clonal expansions (Fig 1B). In cancer (Fig 1C), this means to either *i*) separate tumours from normal cells, *ii*) or detect distinct tumour subclones. Importantly, this deconvolution unravels the composition of a sample (Fig 1D) making it easier to follow up clone-specific downstream analyses (e.g., cell type identification, differential expression or chromatin). By default, CONGAS+ scans for chromosome arms CNAs, but a custom segmentation can also be used; the idea is to search for multimodalities (i.e., peaks) in the RNA/ATAC data distributions. One caveat to detect peaks from RNA compared to ATAC is overdispersion, as transcript counts might be more difficult to separate for peak detection (Fig 1E). The Bayesian model however encodes a customisable categorical prior for discrete CNAs in each segment and cluster, and pools segment-level data to strenghten the signals (Fig 1F). Compared to its predecessor CONGAS, this model leverages a far stronger statistical signal by joining ATAC/RNA to determine discrete CNAs, whereas the original model was continuous, which made it difficult to compare copy number profiles.

The model is a parametric Dirichlet mixture for $K \geq 1$ clusters, determined from counts data $\mathbf{X} = (X^R, X^A)$ of $N \in \mathbb{N}$ RNA and $M \in \mathbb{N}$ ATAC cells, mapped to $I \in \mathbb{N}$ segments (Fig 2). Input $\mathbf{X}$ can be either discrete raw counts, i.e. $X^R \in \mathbb{N}_0^{N \times I}$ and $X^A \in \mathbb{N}_0^{M \times I}$, or continuous values normalised by library size, i.e., $X^R \in \mathbb{R}^{N \times I}$ and $X^A \in \mathbb{R}^{M \times I}$. The distinction among a multiomics CONGAS+ (Fig 2C), and a flat CONGAS+ where cells are independent (Fig 2B) reflects in the model latent variables $\mathbf{z}$ used to determine cell assignments to clusters, summarised from a set of $K$-dimensional mixing proportions $\pi \in \mathbb{R}^K$. For the flat CONGAS+ the $\mathbf{z}$ are independent with dimensions $N$ and $M$ ($\mathbf{z} = (z_R, z_A)$), for the multiomics the $\mathbf{z}$ are shared across RNA and

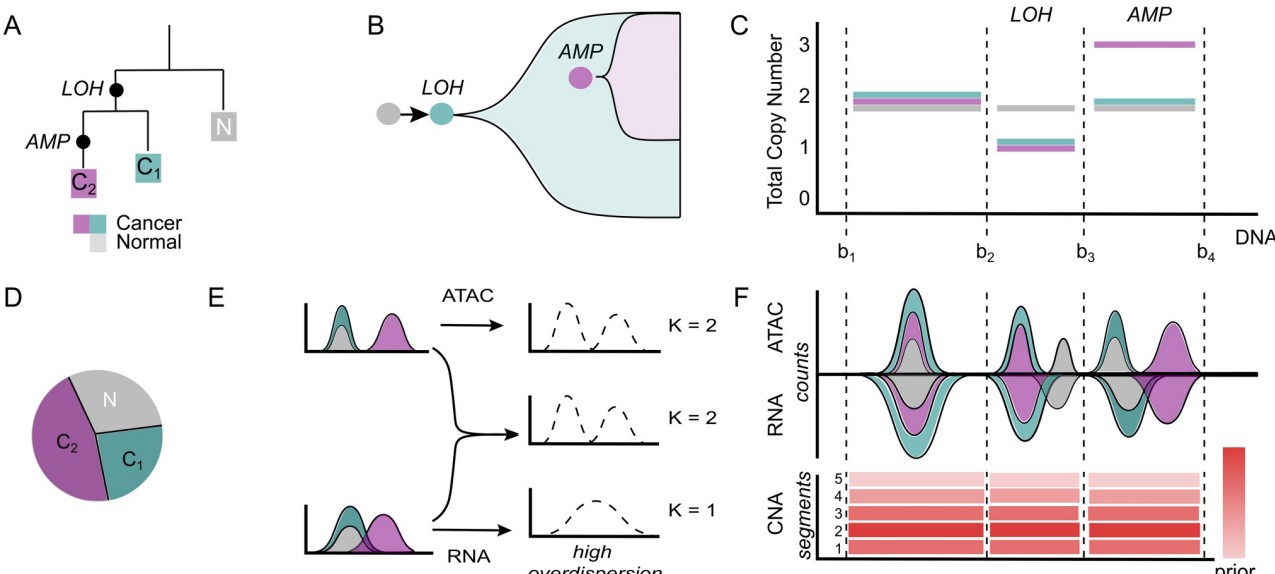

**Fig 1. The CONGAS+ approach. A,B.** Two CNA-associated tumour subclones $C_1$ and $C_2$, evolutionary nested ($C_2$ descend from $C_1$), together with a normal population *N*. Subclone $C_1$ is associated with a loss of heterozygosity (LOH), $C_2$ with an amplification (AMP). **C.** CNA profile for $C_1$, $C_2$ and *N*. In the first segment all the populations are diploid, in the second both $C_1$ and $C_2$ have a single-copy genome and in the third segment $C_2$ has a triploid genome. **D.** Clone proportions in a sequencing assay. **E.** The inference from RNA and ATAC distributions pose different challenges. One of the two (here RNA) might show weaker multimodal signals, making clustering a more challenging task. A joint assay has the advantage to gain the best of the two data types. Here we figure a stronger bimodal signal in ATAC. **F.** On top, cartoon ATAC/RNA signal for each segment. In the first, all clones have similar signals; in the second, normal cells have more ATAC/RNA; in the third, $C_2$ has more ATAC/RNA. RNA signals are more overdispersed, as in panel (E). On bottom, Bayesian categorical priors for the segment values with most mass at 2 (diploid).

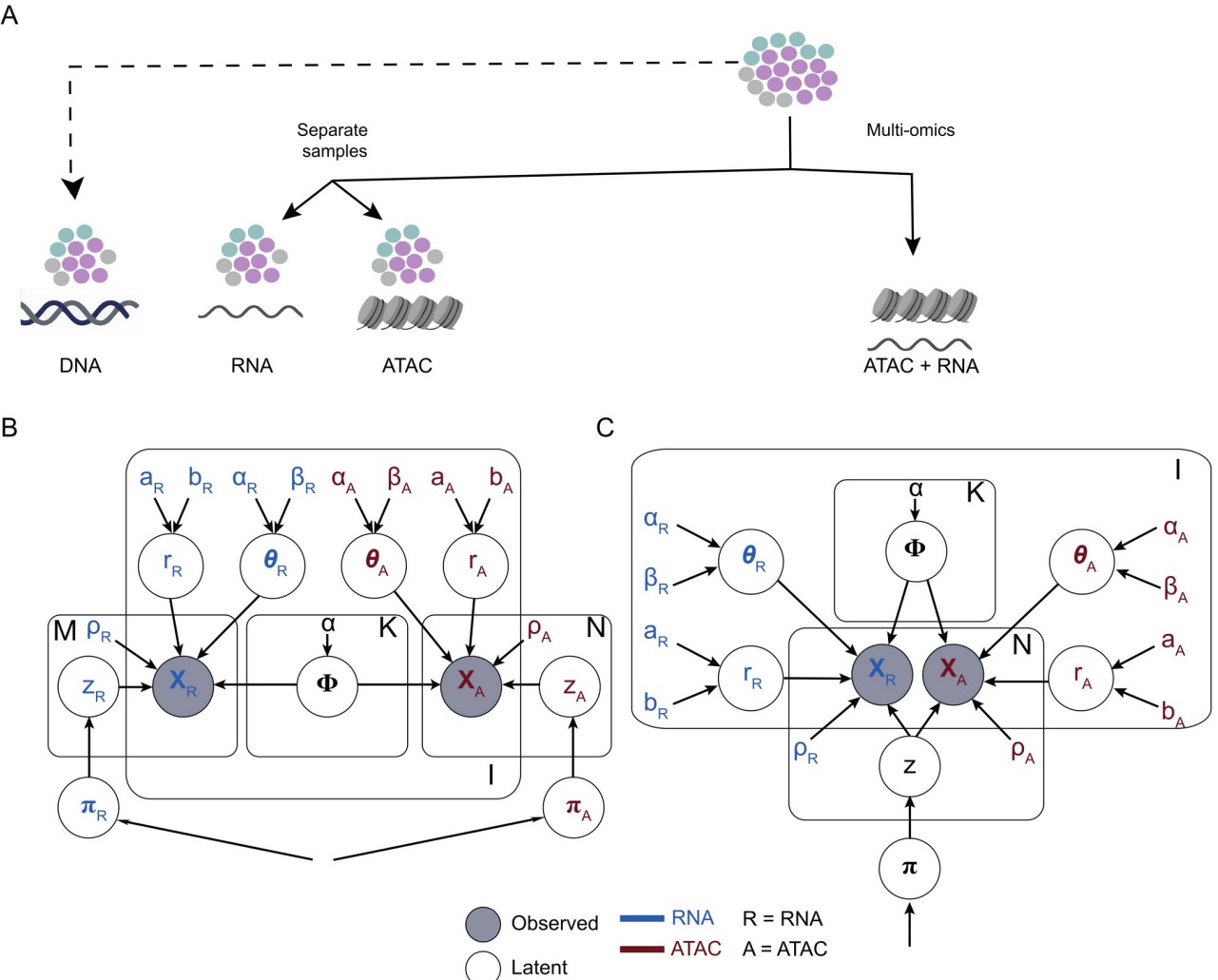

**Fig 2. CONGAS+ graphical models. A** Design of CONGAS+, which can be applied to experimental settings where scRNA-seq and scATAC-seq data are obtained from independent cell splits, or from a multiomics assay (i.e., both measures come from the same set of cells). The input segmentation for CONGAS+ can follow arm-level CNAs, or the profile obtained from an optional bulk sequencing assay. **B, C** Probabilistic graphical models represent observed and latent (i.e., inferred) variables for the flat CONGAS+ (B) and the multiomics (C) extension. Colours encode the ATAC and RNA specific variables, while variables in black are shared. Parameters are learnt via stochastic variational inference in Pyro [26].

ATAC and have a unique dimension $N$. In the flat model it is however possible to share the mixing proportions of among the two types of data, so to ensure that the same proportion of cells in the two data types are assigned to each cluster.

As in other methods [10, 18], the DNA copies of a segment is a linear predictor of $\mathbf{X}$, i.e., if the latent number of DNA copies of a segment $i \in \{1, \ldots, I\}$ is $c$, we predict $r > 0$ RNA transcripts and $a > 0$ chromatin peaks in the segment as $r \propto c\,\theta_i^R$ and $a \propto c\,\theta_i^A$, where each $\theta_i^t \in \mathbb{R}^+$ ($t = R$ or $A$) is a segment-specific rate to observe RNA transcripts and ATAC peaks. The generic CONGAS+ likelihood, for either RNA or ATAC after marginalizing the local latent variable $\mathbf{z}$, is defined as

$$p(\mathbf{X}|\boldsymbol{\theta}, \boldsymbol{\pi}, \boldsymbol{\Phi}) = \prod_{n=1}^{N}\sum_{k=1}^{K}\boldsymbol{\pi_k}\prod_{i=1}^{I}f_k(x_{n,i}|\boldsymbol{\theta_i}, \boldsymbol{\Phi_{k,i}}) \tag{1}$$

where $\pi \in \mathbb{R}^K$ are $K$-dimensional mixing proportions, $x_{n,i}$ the counts for the $n$-th cell and the $i$-th segment, and $\Phi$ is a $K \times I \times H$ tensor for the probability distribution over latent CNAs, which here have $h = \{1, 2, \ldots, H\}$ possible states ($H$ maximum copies). This assumption is justified by the resolution of current single-cell technologies. To map independent scRNA-seq/scATAC-seq on the same set of clusters, $\Phi$ is shared across data types, and $f$ is a generic observational model that depends on $\mathbf{X}$.

For scRNA-seq/scATAC-seq integer counts $x_{n,i}$ mapped to the $i$-th segment of the $n$-th cell, the function $f$ associated with the $k$-th mixture component is a Negative Binomial parameterized by mean and overdispersion, i.e.,

$$f_k(x_{n,i}|\boldsymbol{\theta_i}, \boldsymbol{\Phi_{k,i}}) = \text{NegBin}\left(x_{n,i}\left|\frac{\mu_{k,i,n}}{\mu_{k,i,n} + r_i}, r_i\right.\right) \tag{2}$$

The mean depends on the expected counts per allele ($\theta_i$, learnt from data), the library size factor of the cell ($\rho_n \in \mathbb{R}^+$, observed), and the linear combination (dot product) of the latent CNAs, i.e.,

$$\mu_{k,i,n} = \underbrace{(\rho_n.\theta_i)}_{\text{Normalization}} \cdot \underbrace{\left(\sum_h \Phi_{k,i,h} \cdot h\right)}_{\text{CNA mixture}} \tag{3}$$

where we are using a nested mixture of CNAs with value $h$ weighted by their probability $\Phi_{k,i,h}$—i.e., the probability to detect CNA value $h$, in segment $i$ and cluster $k$. The overdispersion of this segment is instead learnt from data. At the level of priors, the probability for each CNA $\Phi_{k,i,h}$ is a Dirichlet distribution with parameter an $|H|$-dimensional vector $\alpha$, which can be set to any input predefined CNA profile. By default, if we expect $p$ copies for a segment, we assign to the $p$-th entry value 0.6, and to the remaining 0.1 to skew the Dirichlet (the sensitivity to these values is discussed in S1 Text). Let data be $\mathbf{X} = (X^R, X^A)$ the joint RNA/ATAC CONGAS+ log-likelihood has a shrinkage form

$$p(\mathbf{X}|\boldsymbol{\theta}, \boldsymbol{\pi}, \boldsymbol{\Phi}) = \lambda \cdot p(X^R|\underbrace{\boldsymbol{\theta}^R, \boldsymbol{\pi}^R}_{\Omega_R}, \boldsymbol{\Phi}) + (1 - \lambda) \cdot p(X^A|\underbrace{\boldsymbol{\theta}^A, \boldsymbol{\pi}^A}_{\Omega_A}, \boldsymbol{\Phi}) \tag{4}$$

where $0 \leq \lambda \leq 1$ ($\lambda \in \mathbb{R}$) is a fixed hyperparameter to weight the likelihood of both data types, and $\Omega_R$ and $\Omega_A$ are the set of parameters for RNA/ ATAC. Please see S2 Text for the full model formulation. The parameters are learnt via stochastic variational inference in Pyro [26], where we reparameterize the categorical $\Phi$ with a Gumbel-Softmax (S2 Text). The Gumbel-Softmax is a continuous approximation to the discrete distribution, and offers an advantage for stochastic variational inference. This continuous relaxation in fact enables the application of gradient-based optimization such as backpropagation, making the training of models with discrete variables more efficient and scalable, avoiding high-variance samplings from the categorical [27]. The inference returns the full posterior distribution over CNAs, cell clustering assignments and a Maximum A Posteriori (MAP) for all other parameters. The number of clusters $K > 0$ is optimised using standard score functions [28] such as the Bayesian (default) or Akaike Information Criteria (BIC, AIC), as well as the Integrated Completed Likelihood (ICL) [28]. Given the complete log-likelihood

$$L(\mathbf{X}) = \ln p(\mathbf{X}|\theta, \pi, \Phi) \tag{5}$$

and the number of parameters $v$ for a model with $n$ samples, the scores are

$$\begin{aligned}
\text{BIC}(\mathbf{X}) &= v \ln(n) - 2L(\mathbf{X}) \\
\text{AIC}(\mathbf{X}) &= 2\,v - 2L(\mathbf{X}) \\
\text{ICL}(\mathbf{X}) &= \text{BIC}(\mathbf{X}) + H(\mathbf{z})
\end{aligned} \tag{6}$$

where $\mathbf{z}$ are the latent variables for cell assignments, and $H(\mathbf{z})$ their entropy [29]. When the number of data points is greater than the number of model parameters, we use BIC as default for model selection. The full formulation of our model, which is implemented in 2 open-source R and Python packages, allows the user to customise the inference (e.g., select specific score functions or pre-filter segments that contain multi-modal signals).

# Results

## Model validation and parameterization

**Comparison to alternative methods.**   We tested the plain CONGAS+ using synthetic simulations parameterised from two 10x genomics datasets of human peripheral blood mononuclear cells (PBMC) [30, 31]. After quality check and clustering we isolated PBMC monocytes (for scRNA-seq) and neutrophils (for scATAC-seq) using MAESTRO [32], and obtained parameters to simulate RNA (with SPARsim [33]) and ATAC (with simATAC [34]) profiles for putative normal cells. We simulated CNA segments and a clonal architecture for trees with $2 \leq K \leq 10$ clones, adding $n \leq 3$ new segments per clone following the tree structure. We generated clusters mixing proportions from a uniform Dirichlet, and RNA and ATAC counts mapped to CNAs were corrected assuming a linear model. With 10 replicas for each $K$, we created 90 datasets with 1500 scRNA-seq and 1500 scATAC-seq cells each. Please see S3 Text for additional details on synthetic data simulations). We applied CONGAS+ with $K \leq 10$, BIC for model selection and a shrinkage parameter $\lambda = 0.5$, since the same multi-modal signals were simulated for ATAC and RNA. We measured *i)* the Adjusted Rand Index (ARI), i.e. the similarity between the known and inferred cluster memberships, and *ii)* the mean absolute error (MAE) between simulated and inferred CNAs. We reported in Fig 3A–3C the CNAs and low dimensionality representation of ATAC and RNA counts for an example dataset with $K = 3$. For the same cells, Fig 3D and 3E show the count distributions for a segment located on chromosome 3 and 8, characterised by an LOH and an amplification in distinct subclones. CONGAS+ did identify the simulated clones, with density plots mimicking the trends in the data perfectly (Fig 3F and 3G).

Across all tests we observed a very good performance (median ARI > 0.7 and MAE < 1) suggesting that we can retrieve clones assignments and their CNAs (Fig 4A and 4B). We also measured (Fig 4C) run times scaling up to 100.000 cells, observing that CONGAS+ on GPUs is extremely fast.

The same test has been carried out against alternative tools for scRNA-sq or scATAC-seq data (Fig 4A). In particular, we measured ARI for the RNA-based tool copyKAT [19], and the ATAC-based tool copy-scAT [22]. Since these tools require normal cells to infer CNAs, we included a cluster of normal cells in each dataset. For copyKAT, which can only classify normal and tumour cells, we also constructed dendrograms from the inferred copy number matrix, and clustered cells with dynamicTreeCut [35]. The performance of the competing tools was however quite disappointing, and one limitation is the lack of synthetic tests in the corresponding publications that we could use. While we relied on external tools to generate baseline ATAC/RNA signals ([33, 34]), the explicit definition of simulations that can model

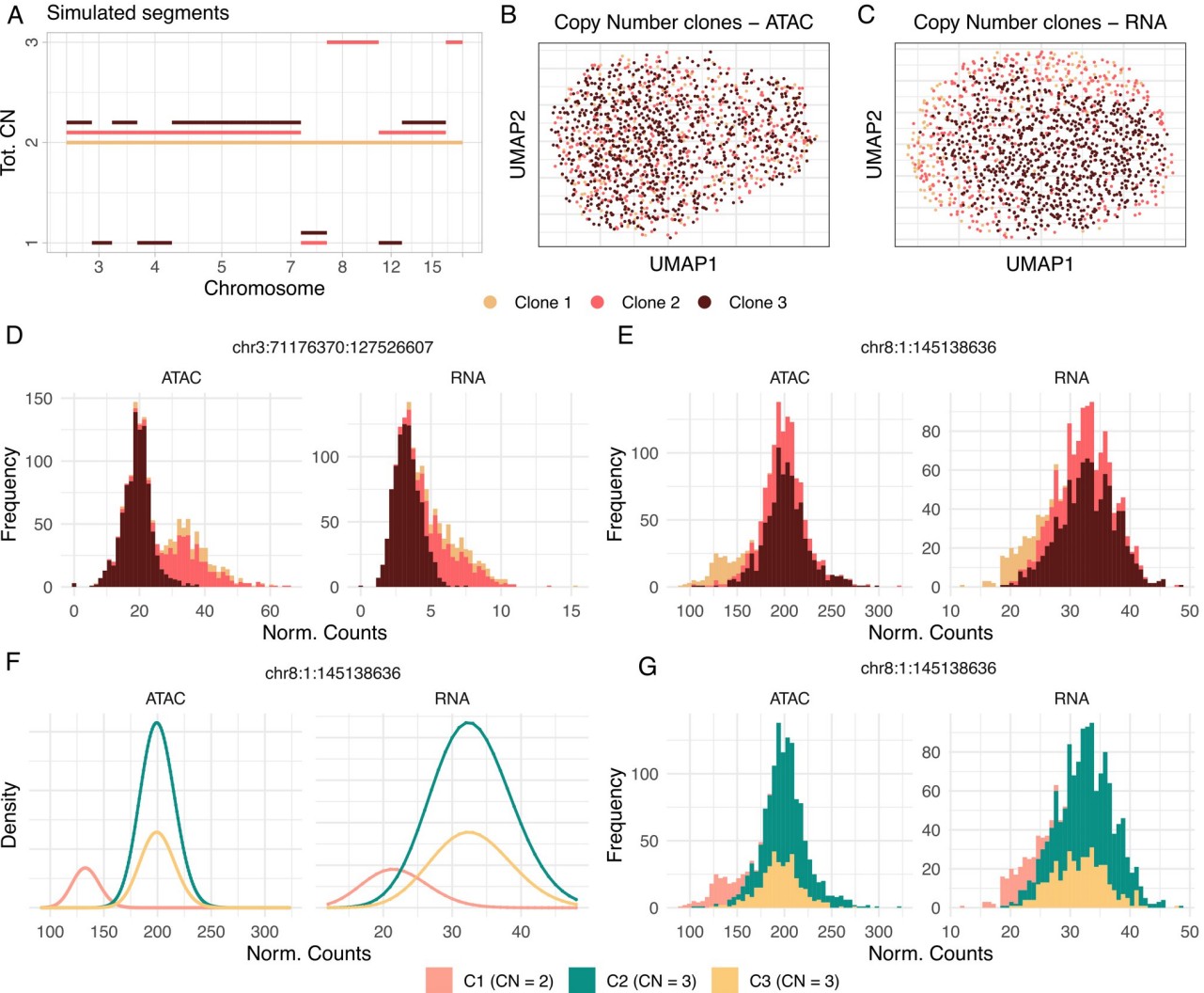

**Fig 3. Simulated scATAC-seq/ scRNA-seq data. A.** Segments breakpoints and copy number values of a synthetic dataset with $K = 3$ clusters. Only chromosomes with one or more CNAs are displayed in the plot. **B,C.** Low dimensional representation of the scATAC-seq and scRNA-seq profiles in panel (A); cells are colored by simulated clone. **D,E.** Data distributions for a segment of chromosome 3, with a loss of heterozygosity in one clone, and for a segment of chromosome 8, with an amplification in a different clone. **F,G.** Probability density functions estimated by `CONGAS+` (F) and data histogram (G) for the chromosome 8 segment in panel (E).

CNAs in scRNA-seq and scATAC-seq is needed to obtain better performance comparison estimates across all these tools.

**Shrinkage to correct uneven ATAC/RNA data quality.** RNA signal quality can be more dispersed (Fig 1E) than ATAC one due to differences in sample preparation, library size, gene expression variability, and sequence-specific biases [36, 37]. Overdispersion is therefore a statistical confounder to separate clusters from scRNA-seq data alone. We selected a dataset with this feature, gathering ∼1800 scRNA-seq and ∼600 scATAC-seq profiles from the Basal Cell Carcinoma (BCC) sample SU008 [38, 39]. We created a dataset of tumour and normal cells in even proportions, restricting the genome to two diploid areas (tumour equal to normal), and two aneuploid (tumour different from normal) areas with bimodal signal poorly evident in RNA (Fig 4D). We bootstrapped the genes in each segment, and compared 30 inferences with

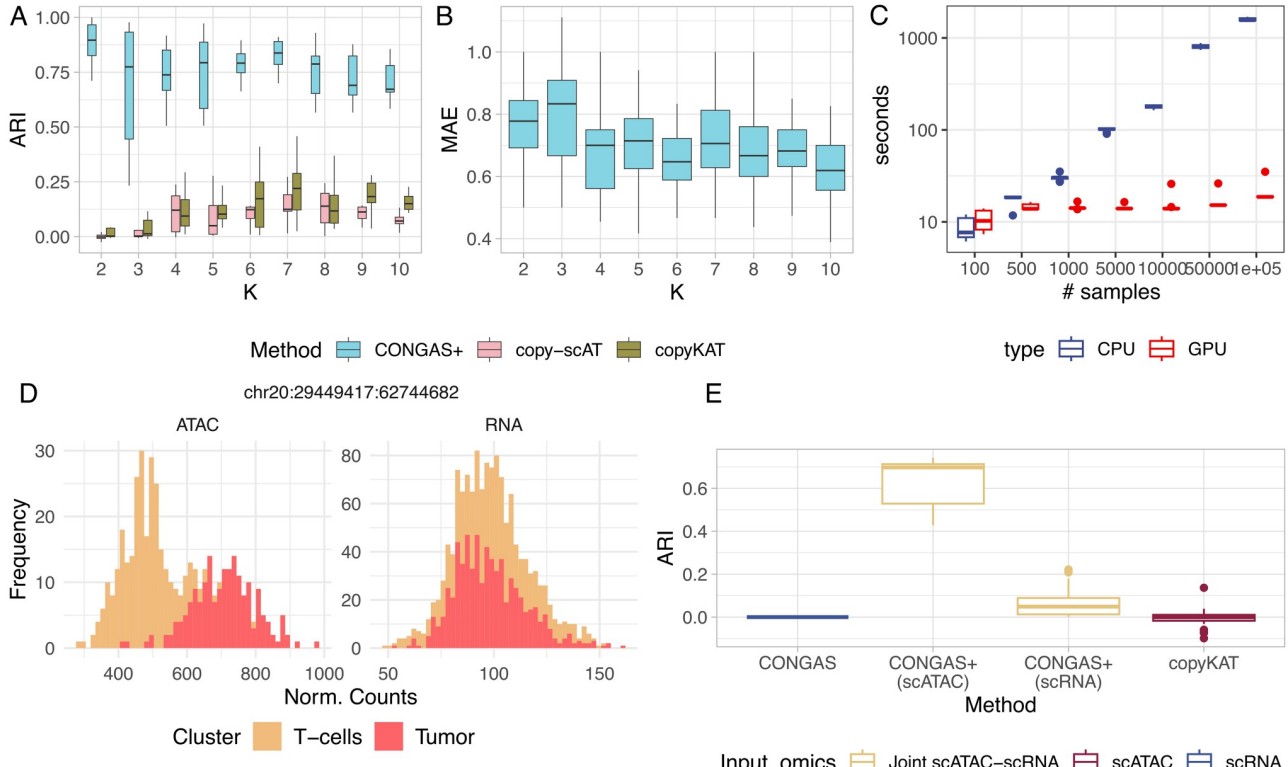

**Fig 4. Results on simulated data. A.** Adjusted Rand Index (ARI) among simulated cells and CONGAS+ clustering assignments, for 90 datasets with 1500 cells for scRNA-seq and 1500 for scATAC-seq. CONGAS+ performance is compared with copyKAT [19] and copy-scAT [22]. **B** Mean Absolute Error (MAE) among simulated and inferred copy number profiles. **C.** Computation times (in seconds) for CONGAS+ with up to 100, 000 cells, on CPU and GPU. **D.** Example counts of a bootstrap sample for an amplified segment with bimodal ATAC signal, and unimodal RNA signal (see Fig 1E). **E.** ARI boxplot for copyKAT, CONGAS and CONGAS+ (computed on scRNA and scATAC separately) in a test simulated as in panel (D).

CONGAS+ (RNA/ATAC), CONGAS (RNA) and copyKAT (RNA). Using a joint ATAC-RNA assay, CONGAS with $\lambda = 0.1$—i.e., weighting 90% the ATAC more than the RNA—detected CNAs that distinguish tumour from normal cells, obtaining a median ARI $\sim 0.7$ on ATAC but a lower ARI on RNA (Fig 4E). In general, due to the weaker RNA signal, all RNA-based tools struggled separating tumour and normal cells, with copyKAT and CONGAS unable to detect the split (Fig 4E), and copy-scAT failing to execute with standard parameters. Overall, this shows that with a joint assay we can detect the latent clone composition even if one of the two data types has weak signal quality.

The shrinkage hyperparameter $\lambda$ can therefore be used to weigh the evidence across data types, to correct for uneven signal quality. This serves as a natural hyperprior, which we optimised—via simulations—to select a default value reasonable for most cases. First, we collected BCC data from samples SU008 and SU006 [38, 39]; SU008 shows a multimodal signal in ATAC while SU006 in both ATAC and RNA. For SU006, we selected two diploid segments and two loss of heterozygosity (LOH) segments (Fig 5F and 5G), and scanned $\lambda = 0.05, 0.15,$ ..., 0.95. Using $K = 2$ (known true value), we performed 10 runs to compare the ARI for cluster assignments against tumour/normal labels [38, 39]. For SU006, we observed RNA/ATAC inferences stable against $\lambda$, with tumour and normal cells always separated (Fig 5C and 5H). For SU008, instead, only ATAC exhibits a neat bimodal distribution (Fig 5B), while we observed (Fig 5C) that for $\lambda < 0.5$ the ATAC ARI is stable at $\approx 0.75$, whereas it decreases as $\lambda \approx 0.95$. Inferences for the best/ worst ARI (Fig 5D and 5E) show discordant tumour and

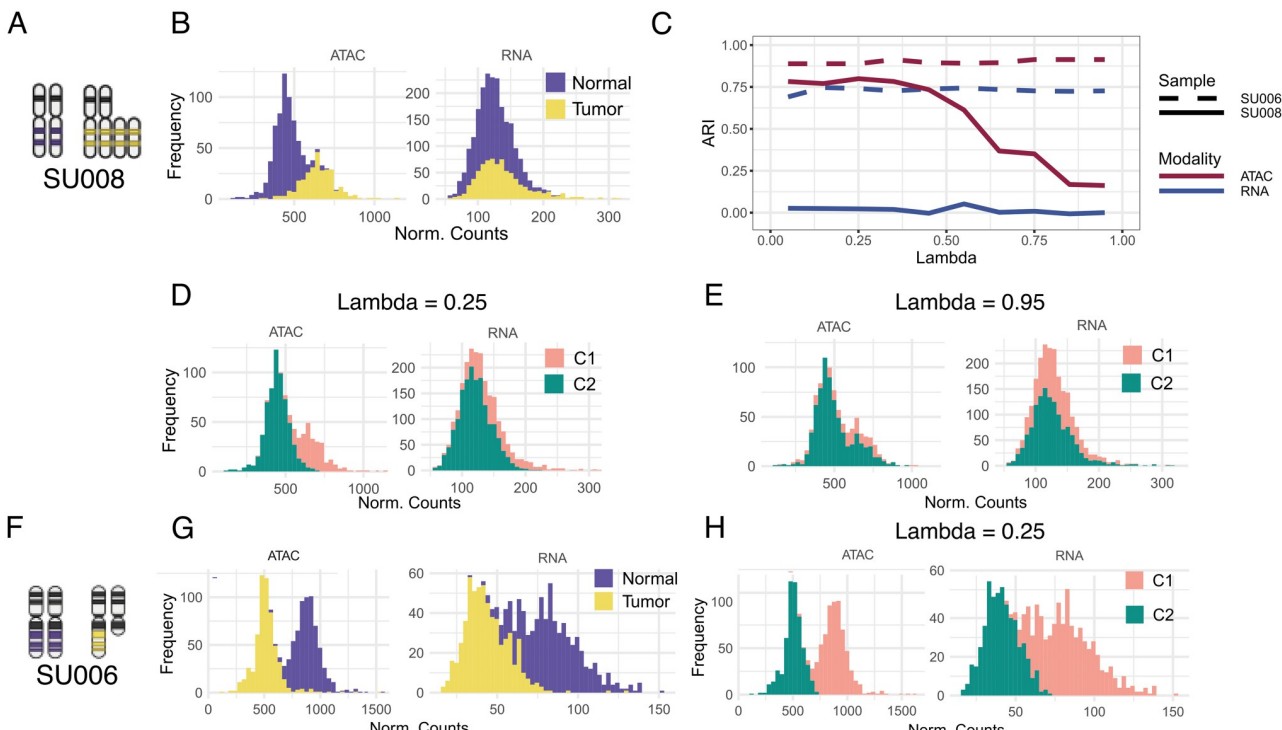

**Fig 5. `CONGAS+` shrinkage effect. A,B.** Segments with bimodal signal (tumour versus normal) in both scRNA-seq and scATAC-seq of the Basal Cell Carcinoma (BCC) sample SU008 [38, 39]. **C.** Adjusted Rand Index (ARI) for `CONGAS+` inference as a function of different values of the shrinkage coefficient λ. Higher values of λ favour RNA over ATAC, and viceversa. The maximum ARI is achieved for low λ and ATAC. **D,E.** ATAC and RNA profiles on the segments in panel (C) show that for low λ cells are split into 2 ATAC clusters. In RNA, instead, regardless of λ the cells can be split, as suggested in Fig 1E. **F-H.** From sample SU006 [38, 39], instead, we obtain a good clustering both in RNA and ATAC data.

normal assignments. With λ ≥ 0.25 `CONGAS+` did not fit the ATAC bimodality, merging 63% of tumour and 90% of normal cells together. Instead, for λ < 0.25—more weight assigned to ATAC—assignments retrieved are perfect. As expected, the model is never able to separate tumour and normal cells from RNA, due to its unimodal distribution. Overall, these tests show that if the quality of ATAC/ RNA are different λ can indeed act as a shrinkage effect, and `CONGAS+` offers a principled approach based on likelihoods to inspect the optimal λ, which is set by default to 0.5. We provide a guideline to tune its value based on the results of the default run, that considers how well the the inferred distributions fit the data in both modalities (See S4 Text).

## Matching RNA and ATAC subclones against single-cell DNA

The most effective technology to detect single-cell CNAs is certainly single-cell DNA sequencing (scDNA-seq) because it directly accesses DNA. This technology is however expensive and laborious, and it is therefore interesting to assess how much we can retrieve from scRNA/scA-TAC sequencing of a sample with matched scDNA-seq because, as alternative technologies, they have the advantage of measuring cellular phenotypes.

For the gastric cancer cell line SNU601, we collected scDNA-seq and scRNA-seq from [40], and scATAC-seq from [20]. In [20], 6 subclones (labeled 1 to 6) were determined from 10 copy number segments using scDNA-seq, and they were matched against scATAC-seq data. From these subclones, we merged clones 3 and 6 (still labeled 6) that were associated with

segments with less than 5 genes. We ran the plain CONGAS+ on scATAC-seq and scRNA-seq using the segmentation and CNA-priors determined from the most abundant scDNA-seq sub-clone in [20] (subclone 2). Our analysis ($\lambda = 0.5$, scored by BIC) identified 4 subclones which we used to verify that CONGAS+ (*i*) maps RNA/ATAC cells to the corresponding scDNA-seq clones and (*ii*) infers the correct integer copy number values for each clone.

First, we computed the overlap between each CONGAS+ cluster and scDNA-seq clone and observed that the majority of cells in each cluster belong to one scDNA-seq clone (Fig 6A). The same cluster-clone correspondence was obtained by the mean absolute deviation between the integer copy number profiles (Fig 6B), which shows that CONGAS+ did retrieve very simi-lar clone assignments and CNAs (full profiles in Fig 6C). ATAC counts projection on a low-dimensional manifold shows that cells in a CONGAS+ cluster are close in the embedding, implying a correlation between genotype and phenotype in the cell line (Fig 6D and 6E). As an example of the clones identified by CONGAS+, in Fig 6F and 6G we show the counts distribu-tions of ATAC clusters on the q-arm of chromosome 11, which are also present in RNA data (Fig 6H).

The advantage of ATAC/RNA over DNA is the possibility of performing phenotype-level differential analyses. We tested for differential gene expression (Wilcoxon test) among CONGAS+ clusters C3 and C4, which shared less markers and therefore seemed more distinct in terms of phenotypes. This analysis revealed processes active in each subclone (Fig 6I), such as the de-regulation of CLDN18, previously reported in gastric cancer [41], and DHRS2, also associated with gastric carcinogenesis [42]. Among the top de-regulated genes (absolute log fold change above 0.5 and Wilcoxon test p-values below 0.01) we found CCNI, which is involved in the induction of angiogenesis and whose up-regulation was correlated with lymph-node metastases in gastric cancer [43]. Moreover, we fond REG4, a gene known to be up-regulated in gastric cancer [44]. At the ATAC level, we identified binding motifs related to differentially expressed ATAC peaks (Fig 6J), identifying Krüppel-like factors, DNA-binding transcriptional regulators with multiple functions regarding proliferation, migration, inflammation and angiogenesis [45], which are essential for tumour development [46]. These results suggest that different mechanisms promote tumour growth in different clones, and our phenotype-level characterisation is a major improvement of a plain DNA-based analysis from scDNA-seq.

## Tumour-normal deconvolution from ATAC/RNA multiomics

Multiomics ATAC/RNA assays represent the ideal test-bed for CONGAS+, as cells are phased against the input measurements. While this type of assay are yet to become common practice, we sought to test our model against B-cell lymphoma data [47] sequenced with the 10x multiome kit [6]. We carried out two types of tests on $\sim 6400$ cells with annotated cell types provided by [47]. One test used the multiomics version of CONGAS+ which shares the latent variables across RNA and ATAC, and the other exploits the plain version where the variables are independent. We aimed at measuring, besides the composition at the subclonal level with the multiomics CONGAS+, if the plain model splits cells consistently with the joint assay.

First, we observed that cell types were distinguishable in a joint ATAC/RNA low-dimen-sional representation [48] (Fig 7A). In particular, we noticed two tumour subclones (B and B-cycling cells), and normal cells split into Monocytes, T, T-cycling and B cells. Biologically, while copy numbers could tell apart normal from tumour cells, the distinction among B and B-cycling cells is more likely linked to cell cycle, a byproduct of transcriptional regulation not necessarily linked to CNAs [49].

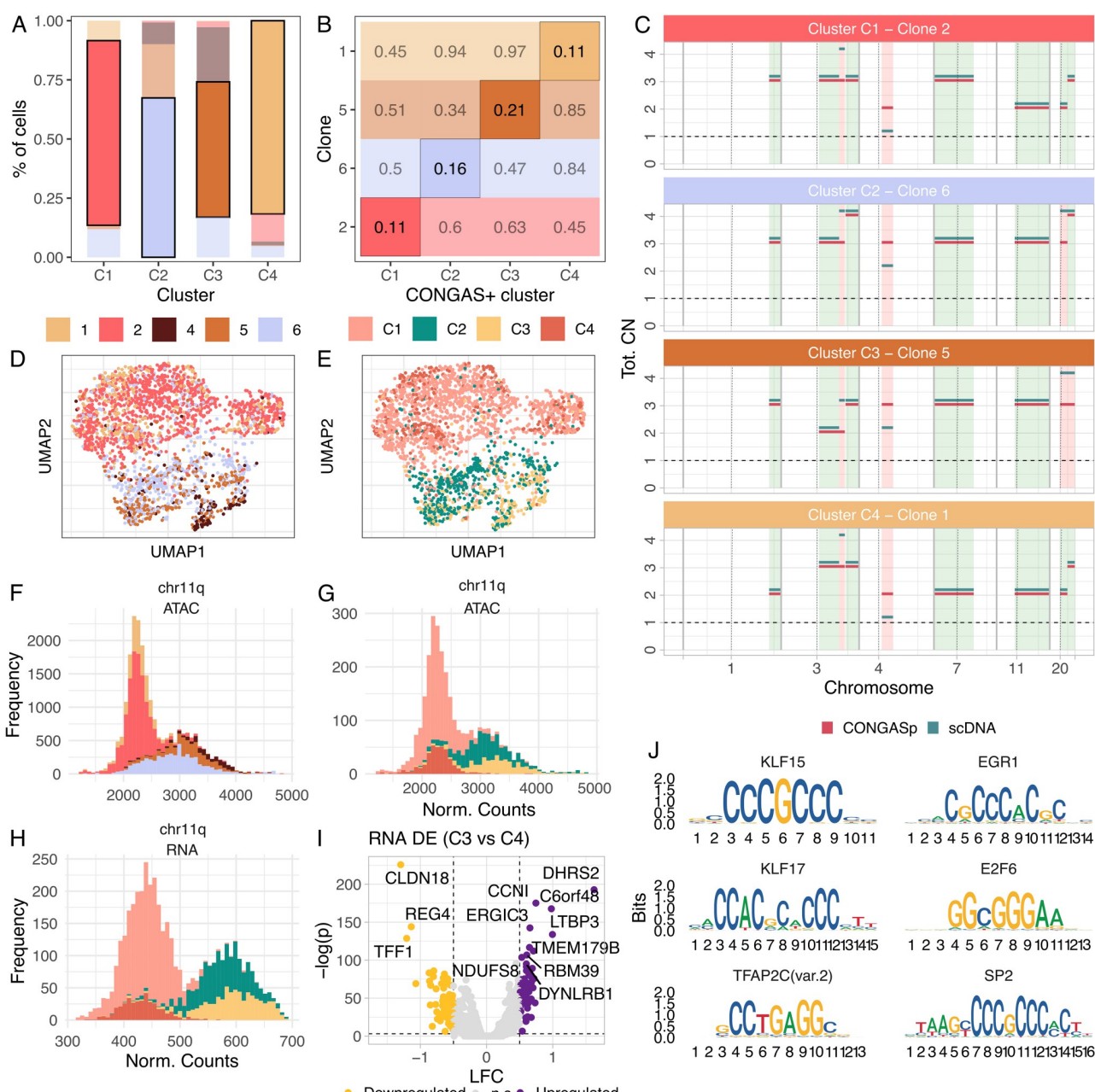

**Fig 6. ATAC/RNA `CONGAS+` analysis versus scDNA-seq. A,B.** Mapping among scDNA-seq clones (ground truth) detected from a gastric cancer cell line (SNU601 [20]), and clusters inferred by CONGAS+ ($\lambda = 0.5$) from independent ATAC/RNA data, using the segmentation of the most prevalent clone from scDNA-seq. The largest cluster per mapping is highlighted to denote that there is almost a one-to-one mapping between the analyses, as is also suggested by the absolute mean absolute deviation between copy number profiles of the two analyses. **C.** CNA profiles for the matched analyses are in large agreement, excluding small segments on chromosomes 3, 4 and 20. **D,E.** A UMAP low-dimensionality representation shows good overlap between analyses. **F,G.** Comparison between the ATAC count distribution on the p-arm of chromosome 20, coloured by ground truth clones and CONGAS+ clusters. **H.** RNA distribution on the p-arm of chromosome 20 as in panels F-G shows concordance among ATAC and RNA. **I,J.** Differential gene expression volcano plot (Wilcoxon test) for two CONGAS+ clusters. and binding motifs associated with differently expressed ATAC peaks in both clusters.

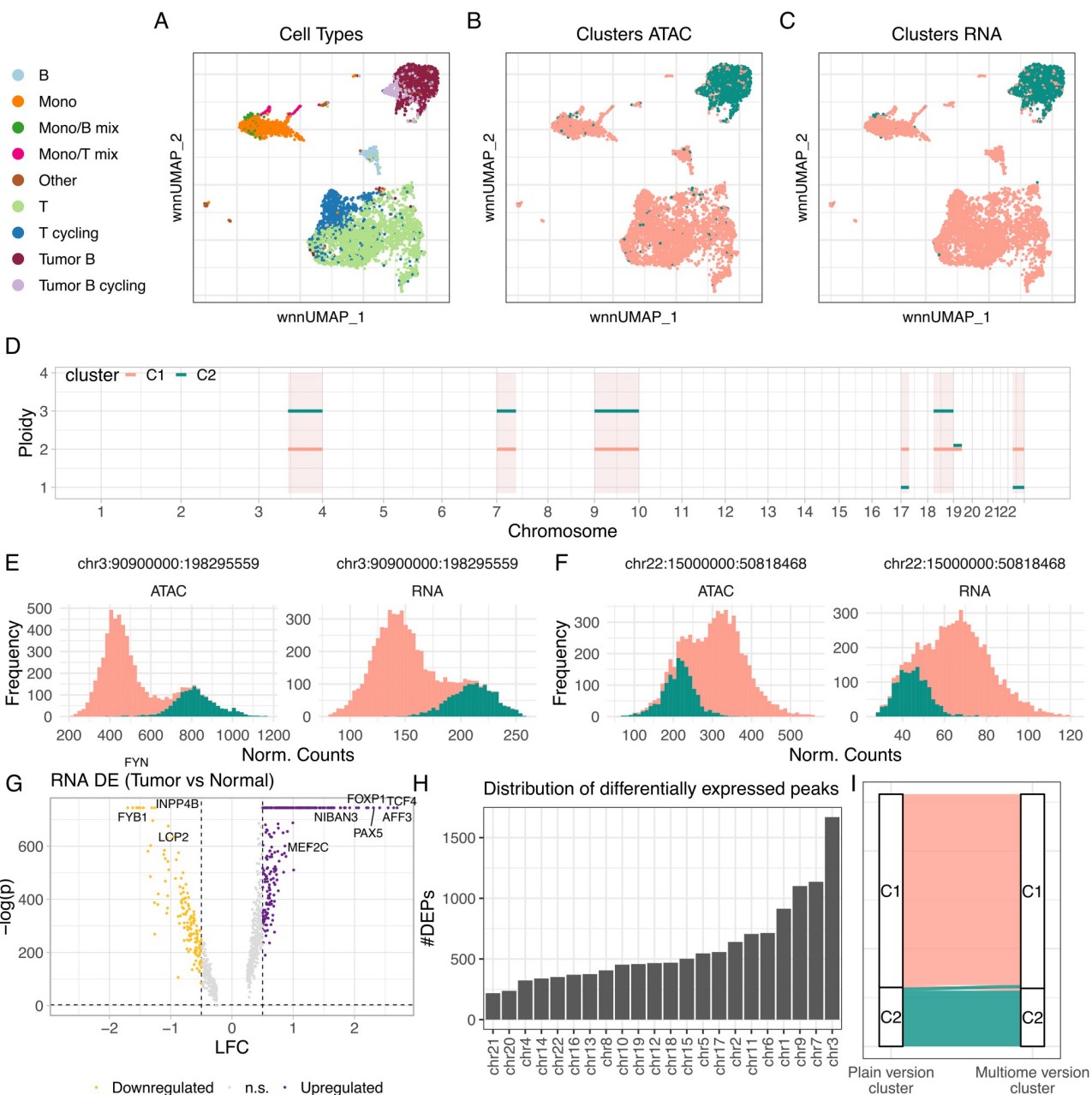

**Fig 7. Application of CONGAS+ to B-cell lymphoma multimodal data. A.** Cell types annotated in a low-dimensionality UMAP representation of ∼6400 RNA and ATAC single-cell data from a 10x multiomics assay [47]. **B,C.** UMAP coloured according to the two clusters inferred by CONGAS+ (multiomics model) from RNA/ATAC data, using an arm-level segmentation. The analysis separates perfectly tumour and normal cells. **D.** Copy number profiles for the two clusters identified by CONGAS+; segments with no lines have the same segments in all clusters. **E-F.** Normalised counts for the q-arm of chromosome 3, where the tumour is amplified, and the p-arm of chromosome 6, where the tumour has a loss. **G-H.** Differential testing for RNA counts (G) and ATAC peaks (H) across the two clusters. **I.** Comparison among clustering assignments the multiomics and flat CONGAS+.

The multiomics CONGAS+ ($\lambda = 0.2$, scored by BIC) splits tumour and normal cells (Fig 7B and 7C) with $K = 2$ with arm-level segments. As Bayesian priors, we specified vague diploid values with slightly higher concentration mass for the diploid cluster, which encoded our knowledge of normal cells in the assay. The profiles inferred (Fig 7D) for normal cells are

consistently diploid, while the tumour have an amplification (trisomy) on the p-arm of chromosomes 7, 9, and the q-arm of chromosomes 3, 9, 18. Moreover, tumour cells have a loss of one allele, and henceforth of heterozygosity, in the p-arm of chromosome 17 and the q-arm of chromosome 22. In all other chromosomes tumour and normal cells are diploid heterozygous. The statistical signals of some of these chromosomes were quite unequivocal (Fig 7E and 7F).

We performed differential analysis for both RNA and ATAC, comparing tumour and normal cells for RNA transcripts (Fig 7G) and open chromatin peaks (Fig 7H). We identified, from RNA, differences in the expression of genes that distinguish normal from lymphoma cells (absolute log fold change above 0.5 and Wilcoxon test p-values below 0.01). The strongest signals were for INPP4B, a tumour suppressor gene [50] which we find down-regulated in tumour cells, LCP2, a prognostic gene for metastatic melanoma-infiltrating CD8+ T cells [51], MEF2C, a gene linked with lymphoma pathogenesis [52], and FOXP1, a gene that has been linked with worse survival in of diffuse large B-cell lymphomas [53]. From ATAC, we found most signals on chromosomes 3, 7 and 9 Fig 7H, which are also affected by CNAs. We ranked chromosomes by deferentially-opened peaks and find also chromosomes 1, 6 and 11 immediately following the segments linked with CNAs, suggesting differences in chromatin accessibility not directly linked to copy number events. Interestingly, clustering assignments of multiomics and plain CONGAS+ were consistent (Fig 7I), suggesting that even when we do not treat cells as paired observations, we can still retrieve good clone assignments. Overall, even if we did not detect CNA-associated tumour subclones in this lymphoma, this case study shows the combined power of ATAC and RNA, a joint assay that, to the best of our extent, has not yet been exploited before to study CNAs in single cells.

## Drug-resistance associated with copy number subclones

Phenotypes associated with CNAs and chromosomal instability can sometimes confer resistance against anticancer drugs [54]. Using data of the prostate cancer cell line LNCaP generated from [55], we tested if CONGAS+ could identify CNA-associated subclones in a drug-screening assay. In the experimental design of [55], single-cell ATAC/RNA data were generated for LNCaP (parental), and then for one line treated 48 hours with androgen receptor antagonist enzalutamide (ENZ), and two resistant lines (RES-A and RES-B) derived after long-term exposure to ENZ and diarylthiohydantoin RD-162, respectively (Fig 8A). To define segments and priors we used LNCaP cytogenetics data from the DepMap portal [56]. Then we merged the 4 samples (parental, ENZ-48, RES-A, RES-B), and filtered out segments with more than 10% of cells showing zero counts.

The plain CONGAS+ ($\lambda = 0.5$, scored by BIC) identified 5 clusters, with parental and ENZ-48 cell lines clustered together, and the two resistant cell lines split in 4 clusters Fig 8B. This is consistent with the experimental design: ENZ-48 has not yet acquired resistance due to its short-term exposure to ENZ and clusters with parental cells, whereas cells from the other two lines are fully resistant thanks to multiple CNAs. CONGAS+ inferred CNAs private to each resistant line, such as an amplification on the p-arm of chromosome 1 (Fig 8D), as well as an amplification on the p-arm of chromosome 6 that is private to clusters C3 and C5 (Fig 8E). Interestingly, among the inferred clusters C3 and C5 have the same CNAs, apart from an amplification on arm p of chromsome 1 (private to C3). We performed differential expression testing to found that cluster C3 over-expresses ENO1, BNIP3, PGK1 and LDHA, 4 genes associated with hypoxia (absolute log fold change above 0.5 and Wilcoxon test p-values below 0.01), that have been previously associated with cancer progression. From the same analysis comparing sensitive cells in C1 to resistant cells in the remaining clusters (Fig 8F), we detected NEAT1 up-regulation, a gene that promotes gastric cancer angiogenesis [57]. We also

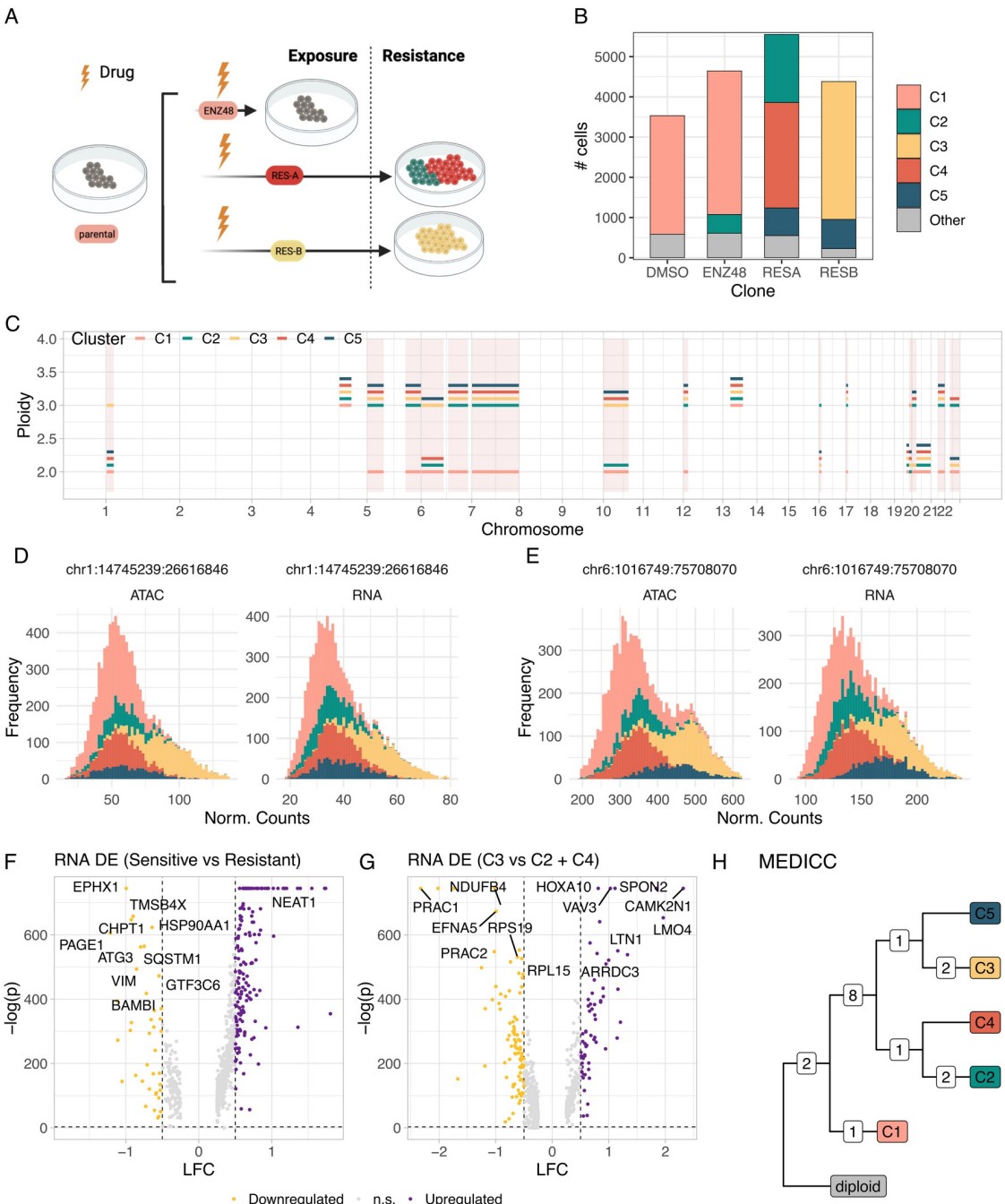

**Fig 8. CNA-associated drug resistance with `CONGAS+`.** CONGAS+ application to a prostate cancer dataset from [55], composed of a mixture of four cell lines with 7600 scRNA-seq cells and 8800 scATAC-seq cells. **A.** Cartoon representing the design of the drug resistance experiment. **B.** Distribution of the 5 clusters inferred by CONGAS+ across the original cell lines. **C.** copy number profiles inferred by CONGAS+ for each cluster. **D,E.** density plot and histogram of normalised counts coloured according to cluster assignments for chromosome 1p (D) where an amplification event is private to cluster C3, and chromosome 6p (E) where an amplification is shared by clusters C3 and C5. **F,G.** Volcano plot showing differentially expressed genes between C1 and the rest of the cells (F) and between cells in C3 and cells in C2 and C4. **H.** Phylogenetic tree inferred with `MEDICC2` [60], using CNAs inferred by CONGAS+. Tips are labeled according to the inferred cluster, and edges are labeled with the number of events that accumulate in the corresponding branch.

compared the resistant subclone C3 to C2 and C4 (Fig 8G), identifying LMO4 upregulated in C3, a marker of cell proliferation and invasion [58], and EFNA5 up-regulated in C2 and C4, a gene regulating cancer cells invasion [59]. As in other datasets, this suggests different mechanisms of tumour progression active in different clones.

Finally, we performed a phylogenetic tree reconstruction among the 5 clusters using MED-ICC2 [60]. The evolutionary relationships between clones (Fig 8H), and the number of CNAs that distinguish parent from child nodes, reveals the expected architecture. Cluster C1 (sensitive cells) is the closest—in terms of evolutionary distance—to the root, and has 3 copy number events, two of which are shared with all other clusters. Next, we find that 8 additional alterations are shared among resistant clones, and then there is a subset of CNAs which distinguish between the different subclones identified by CONGAS+. Overall, this analysis shows that lineage relations associated with CNA-associated subclones can be effectively detected by CONGAS+ and longitudinal data, posing the bases for more systematic investigations on the causal roles of CNAs in promoting therapy resistance.

## Discussion

In cancer, the relation between genotypes and phenotypes is extremely complex and intimately related to the underlying evolutionary dynamics of cancer cells and the environment. To understand this mapping, single-cell technologies can be adopted to achieve a fine-grained resolution of the measurements, but methods are required to resolve signals in high-dimensional noisy data. In this paper, we approached this problem from RNA and ATAC single-cell sequencing, inferring latent tumour subclones associated with CNAs, a specific type of complex genomic mutation that can determine the outcome of cancer dynamics. CONGAS+ is the first Bayesian model that can jointly analyse RNA and ATAC to infer CNAs while clustering cells, and contains a formulation that explicitly supports multiomics assays. The model, in general, has a shrinkage formulation to weigh the evidence between the two data types, a feature motivated by our experience. In fact, in real-world analyses, we often observed scRNA-seq associated with quite dispersed signal, a phenomenon that cannot be fully resolved by just using negative binomial distributions with overdispersion. While we do not have a final explanation on the difference among the data types, one consideration could be that ATAC is a direct measurement of DNA, and therefore is more faithful to genome copy numbers, while RNA is a byproduct of genome composition. Regardless, the tool is designed to support custom parametrisation that can be used to get the best of the data at hand.

Using simulations, in fact, we assessed that CONGAS+ is robust and accurate in retrieving both the clonal composition and corresponding CNAs. This was further confirmed with real data, where the method was able to extract evolutionary relations that are difficult to retrieve with tools that analyse just RNA or ATAC. Moreover, our model could also run seamlessly on multiomics data where RNA and ATAC are measured from the same cell, a scenario that will likely become more common in the future. In general, the deployment of GPUs also allowed to analyse over 100.000 cells in matters of seconds, a feature that will become crucial when datasets become larger.

Following our stream of works, as next steps we plan to introduce further data types, e.g., methylations [61], as well as finer-resolution information such as B-Allelic frequencies and depth ratios commonly used to detect CNAs from bulk [62]. This would allow one to infer allele-specific CNAs, opening also the possibility to perform a de novo genome segmentation of the input single-cell data, lifting the current limitation of CONGAS+ which requires a predefined input genome segmentation.

## Supporting information

**S1 Text. Inference robustness against priors.**
(PDF)

**S2 Text. Extended version of materials and methods.**
(PDF)

**S3 Text. Additional details on simulations.**
(PDF)

**S4 Text. Tuning the hyperparameter $\lambda$.**
(PDF)

## Author Contributions

**Conceptualization:** Giulio Caravagna.

**Data curation:** Lucrezia Patruno, Salvatore Milite.

**Formal analysis:** Lucrezia Patruno, Salvatore Milite, Nicola Calonaci.

**Funding acquisition:** Marco Antoniotti, Alex Graudenzi, Giulio Caravagna.

**Investigation:** Lucrezia Patruno, Salvatore Milite.

**Methodology:** Lucrezia Patruno, Salvatore Milite, Riccardo Bergamin.

**Project administration:** Giulio Caravagna.

**Resources:** Marco Antoniotti, Alex Graudenzi, Giulio Caravagna.

**Software:** Lucrezia Patruno, Salvatore Milite.

**Supervision:** Giulio Caravagna.

**Validation:** Lucrezia Patruno, Salvatore Milite, Riccardo Bergamin, Nicola Calonaci.

**Visualization:** Lucrezia Patruno, Salvatore Milite, Nicola Calonaci, Alex Graudenzi.

**Writing – original draft:** Lucrezia Patruno, Salvatore Milite, Riccardo Bergamin.

**Writing – review & editing:** Lucrezia Patruno, Salvatore Milite, Alberto D'Onofrio, Fabio Anselmi, Marco Antoniotti, Alex Graudenzi, Giulio Caravagna.

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
