## [Decision Letter · Decision Letter 0]

30 Sep 2023

Dear Dr Caravagna,

We are pleased to inform you that your manuscript 'A Bayesian method to infer copy number clones from single-cell RNA and ATAC sequencing' has been provisionally accepted for publication in PLOS Computational Biology.

Best regards,

Teresa M. Przytycka

Academic Editor

PLOS Computational Biology

Jian Ma

Section Editor

PLOS Computational Biology

Reviewer's Responses to Questions

**Comments to the Authors:**

Reviewer #1: The authors should be commended for the amount of new work put in place to address reviewers comments, with new analysis and changes that now greatly improved the manuscript. The authors have addressed all my concerns satisfactorily and the manuscript is now ready to be considered for publication in the journal.

Reviewer #2: The authors addressed all issues raised be the reviewers.

**Have the authors made all data and (if applicable) computational code underlying the findings in their manuscript fully available?**

Reviewer #1: Yes

Reviewer #2: Yes

PLOS authors have the option to publish the peer review history of their article (what does this mean?). If published, this will include your full peer review and any attached files.

Reviewer #1: No

Reviewer #2: No

---

## [Editor Report · Acceptance letter]

26 Oct 2023

PCOMPBIOL-D-23-01275 

A Bayesian method to infer copy number clones from single-cell RNA and ATAC sequencing

Dear Dr Caravagna,

I am pleased to inform you that your manuscript has been formally accepted for publication in PLOS Computational Biology. Your manuscript is now with our production department and you will be notified of the publication date in due course.

With kind regards,

Dorothy Lannert
